# Brain Health Indicators Following Acute Neuro-Exergaming: Biomarker and Cognition in Mild Cognitive Impairment (MCI) after Pedal-n-Play (iPACES)

**DOI:** 10.3390/brainsci13060844

**Published:** 2023-05-23

**Authors:** Kartik Nath, IreLee Ferguson, Alexa Puleio, Kathryn Wall, Jessica Stark, Sean Clark, Craig Story, Brian Cohen, Cay Anderson-Hanley

**Affiliations:** 1Union College, 807 Union Street, Schenectady, NY 12308, USA; 2Gordon College, 255 Grapevine Rd, Wenham, MA 01984, USA; 3iPACES LLC, 56 Clifton Country Road, Suite 104 (Box#11), Clifton Park, NY 12065, USA

**Keywords:** MCI, salivary biomarkers, executive function, exergaming, Alzheimer’s disease

## Abstract

Facing an unrelenting rise in dementia cases worldwide, researchers are exploring non-pharmacological ways to ameliorate cognitive decline in later life. Twenty older adults completed assessments before and after a single bout of interactive physical and cognitive exercise, by playing a neuro-exergame that required pedaling and steering to control progress in a tablet-based video game tailored to impact executive function (the interactive Physical and Cognitive Exercise System; iPACES v2). This study explored the cognitive and biomarker outcomes for participants with mild cognitive impairment (MCI) and normative older adults after 20 min of pedal-to-play exercise. Neuropsychological and salivary assessments were performed pre- and post-exercise to assess the impact. Repeated-measures ANOVAs revealed significant interaction effects, with MCI participants experiencing greater changes in executive function and alpha-amylase levels than normative older adults; within-group changes were also significant. This study provides further data regarding cognitive effects and potential mechanisms of action for exercise as an intervention for MCI.

## 1. Introduction

### 1.1. Dementia Epidemic and Mild Cognitive Impairment (MCI)

Aging can be characterized by cognitive changes such as executive function deterioration; however, these changes can occur earlier in life in those who develop Alzheimer’s disease (AD) and related dementias [1]. There is a dementia epidemic that poses a serious risk to the older population, as the incidence of dementia is expected to reach 152.8 million by 2050 [2]. This is concerning for the older adult population because there are serious limitations in the clinical treatment of Alzheimer’s and dementia, as there is no specific cure for the disorder, and pharmacological treatments have limited effects on dementia progression [3,4,5]. Therefore, researchers worldwide must focus on the development of non-pharmacological treatments, such as exercise, to delay the progression of cognitive decline toward dementia. One possible way to delay the onset of cognitive decline toward dementia [6] is to intervene early, in the prodromal phase of mild cognitive impairment (MCI), to ameliorate deficits in affected cognitive domains. The potential to delay the onset of dementia could reduce the prevalence of dementia [7]; a delay by one year could prevent close to 10 million cases of AD by 2050 [8].

While behavioral interventions may not reverse cognitive decline, there is evidence that physical activity can reduce the risk of dementia onset [9,10,11] or slow progression [7,12]. Unfortunately, many older adults do not exercise [13]. Therefore, researchers are developing innovative methods to engage older adults in physical exercise, such as virtual reality or video games that incorporate exercise (exergames; [14]), as well as exergames that may help older adults to reap the benefits of exercise by prospectively targeting neuropsychological functions such as executive function (neuro-exergame; [15]). A recent review shows that a behavioral intervention consisting of physical and mental exercise has the potential to improve executive control and global cognition in MCI patients [16]. This present study aims to replicate and extend a prior finding that a single bout of interactive neuro-exergaming, an integrated physical and cognitive exercise intervention, could produce enhanced cognitive benefits in the executive function domain [17]. Moreover, this study measured changes in biomarkers that are associated with cognitive improvements [17]. 

### 1.2. Physical Exercise and Cognition Change

Recent and prior meta-analyses [12,18,19,20] have found evidence demonstrating that physical activity can improve cognitive function in later life. More specifically, some studies have found that physical exercise can offer small benefits to executive function in older adults [21,22,23,24]. However, recent reviews have found little evidence in the relationship between physical exercise and cognitive benefits in the literature [25,26]. Nevertheless, some reviews have explored the effect of physical exercise on neurocognitive disorders with the goal of improving cognitive function [18,27,28,29]. Studies have explored the use of physical exercise to achieve cognitive improvements in MCI individuals [30,31,32]. There is evidence that supports the notion that physical activity benefits cognition, especially in the executive function domain, among those in the MCI population [20,33,34,35]. Recent reviews strongly encourage larger randomized controlled trials (RCTs) in the MCI population, where physical activity interventions can explore the broader neuropathology of cognitive decline [16,36,37]. 

Researchers have used different cognitive and exercise interventions to study the effects of exercise in the older adult population [12]. The current study focuses on the MCI population as opposed to more progressed forms of Alzheimer’s disease and related dementias, because the activation of the exercise-induced neurobiological mechanisms has led to greater cognitive improvements in the MCI population [4,38,39]. Many studies have shown that exercise benefits cognition—for example, in the executive function domain, in MCI patients [30,31,32,33,35,40]. Compared to various exercise interventions, aerobic exercise has maintained a medium effect size on global cognition in the MCI population [34], and acute aerobic exercise can benefit executive function in the older adult population [41] Ultimately, physical exercise has been shown to be a reliable method to bring about cognitive improvements in the Alzheimer’s and MCI population [42]. 

Given that most older adults do not exercise at adequate levels, researchers have used exergaming as a motivation for exercise, hoping that an engaging exergame would stimulate consistent exercise toward proper dosing to maximize the cognitive benefits of physical exercise [43]. It was hypothesized that over three months, older adults pedaling a stationary bike along a virtual reality pathway [43] would experience greater cognitive benefits than those who were assigned to the traditional stationary bike. Analyses showed that the cybercycle condition yielded cognitive benefits as hypothesized, as the cybercycle condition yielded a 23% decreased risk of developing MCI [43]. In the end, it was found that the improvements were not due to the dose of exercise: it appeared that the combined physical and cognitive exercise yielded a synergistic effect on cognition, leading to enhanced improvements [43]. The interactive Physical and Cognitive Exercise System (iPACES v1) was developed in an attempt to maximize the neuropsychological benefit that might be derived from combined mental and physical exercise, not simply utilizing a “dual-task” paradigm (e.g., pedaling while playing a separate, non-interactive game) but rather emphasizing interactivity, where pedaling controls on-screen movement and where steering is involved in intentionally designed mental processing tasks that also adaptively impact the physical exercise, i.e., a neuro-exergame [44].

Current theories suggest that the cognitive benefits of physical exercise can be attributed to multiple neurobiological mechanisms that promote neuroplasticity and neurogenesis in the brain [45]. Prior animal research showed that physical exercise and mental exercise activate different neurobiological mechanisms, as physical exercise promotes neurogenesis and mental exercise promotes neuronal survival [46,47,48]. Additionally, human research has shown that physical and mental exercise induce different structural and functional differences in the brain [49]. Studies have revealed various biomarkers correlated with changes in cognition that are indicative of the broader physiological mechanisms [50]. One biomarker, Brain-Derived Neurotrophic Factor (BDNF), is hypothesized to be activated by physical exercise, which is associated with blood flow improvements and neuronal growth [51]. The current iPACES study intended to investigate the enhanced cognitive benefits of interactive cognitive and physical exercise and measure certain biomarkers that may be associated with the underlying neurological biochemical and cellular changes associated with cognitive decline.

### 1.3. Cognitive Training and Cognitive Decline

The surprising findings of the cybercycle study [43] have influenced researchers to investigate the possibility that the cognitive training component has some synergistic effect with physical exercise. Despite our own skepticism, recent guidelines from the American Academy of Neurology [52] state that clinicians should recommend cognitive interventions over pharmacological interventions for MCI patients [52]. Prior meta-analyses indicate that cognitive training interventions do produce effects in the older adult population [17,53,54,55,56,57]. Additionally, prior studies have shown that cognitive interventions alone, such as interactive computer tasks, yield cognitive improvements in older adults [58,59,60,61].

However, researchers have raised concerns regarding the validity of the literature concerning cognitive interventions. One Cochrane review of 11 RCTs revealed no significant benefits from cognitive training [62]. Critics question the cognitive training intervention literature regarding the transfer and generalization of effects to overall cognitive functioning and everyday tasks from cognitive training tasks [32,63,64]. A review of the literature on cognitive training devices found that studies reported significant findings with several limitations, weakening the validity of the tests [56,57,65,66,67]. Furthermore, a critique of the literature has found further limitations, including reporting, generalizability, and more, that weaken the enthusiasm surrounding the findings [62,68,69,70,71,72]. Given the controversy regarding the cognitive benefits resulting from cognitive training alone, it is possible that one can reap the benefits of cognitive training when it is integrated with other interventions, such as physical activity, because of the activation of additional neurobiological mechanisms. Thus, researchers are investigating the benefits of interventions that integrate cognitive training with interventions such as physical exercise.

### 1.4. Mental and Physical Exercise Combination (Interactive/Neuro-Exergaming) and Cognitive Decline

Research has recently begun to understand the interaction between physical and mental exercise, specifically the physiological mechanisms induced by physical exercise that lead to greater cognitive improvements. When physical exercise is combined with cognitive training, the resulting impact may allow the brain to experience greater cognitive benefits. Many reviews have shown the added cognitive benefits of interventions where physical exercise is combined with mental exercise [73,74,75,76,77,78,79,80,81]. Many theories have offered neurobiological explanations for the combined effects of physical and exercise training [77]; however, more technological innovations are needed to explore the physiological impact and effects, specifically in the MCI population [82].

There is a vast amount of literature that covers the cognitive effects of different types of multimodal interventions: (1) combined interventions, where physical and cognitive exercise components are administered sequentially [57,79,83,84,85,86,87]; (2) dual-task interventions, where the components are administered as separate types of tasks at the same time [88,89,90,91,92,93,94,95]; and (3) interactive interventions, a specialized form of exergaming, in which the actions in one component directly affect the other synergistically [15,17,44,96,97,98,99,100,101,102,103].

The theory that integrated (mental + physical exercise) cognitive interventions slow the decline in cognition in older adults with MCI has led to an increase in RCTs. Meta-analyses and reviews explore the synergistic effects of different combinations of mental and physical exercise [76,82,104,105]. Combined interventions have yielded cognitive benefits in the older adult population, specifically in the domains of executive function and memory [34,35,46,48,49,53,54], as well as in global cognition [59]. A recent meta-analysis looked at four studies that included MCI samples and reported that combined interventions were more effective than dual-task interventions [104]. However, some studies have shown that when combined interventions are compared to physical and cognitive interventions alone, they do not produce significantly greater cognitive improvements [78,85,88,100,106].

Dual-task or simultaneous interventions have led to improved cognition in older adults with a variety of exercise activities [82,91,94,107,108,109,110,111]. A recent meta-analysis revealed that integrated physical and cognitive exercise interventions produced significant improvements in cognitive performance, and simultaneous physical and cognitive interventions were superior to combined interventions [73]. Recent reviews of the literature on the cognitive benefits of interactive exergaming have shown that there is evidence of a cognitive improvement [15,76,98,99,112,113,114,115,116]. There remains little research that compares the cognitive and neurobiological outcomes of simultaneous/interactive and sequential interventions, but it is still hypothesized that interactive interventions have a synergistic effect [15,17,43,44,82,102,103,117]. 

Researchers evaluated the effects of various interactive physical and mental exercise combinations in a recent six-month RCT called the Aerobic and Cognitive Exercise Study (ACES) for MCI [44]. Participants were assigned to three conditions: (1) exer-tour (pedaling along a virtual scenic bike path; (2) exer-score (pedaling through an interactive game involving dragons and coins; (3) game-only (no pedaling). Analyses revealed an increase in executive function after 6 months of exercising in both the exer-tour and exer-score samples [44]. However, the study had several limitations: the lack of portability of the commercial grade equipment, which made the intervention experience difficult for the MCI participants; the lack of flexibility in the game to adapt to various cognitive abilities and target specific cognitive domains; and the lack of a reliable method to measure “mental exercise” behaviors [44]. To overcome these limitations, Anderson-Hanley et al. developed a portable neuro-exergame, iPACES v1, that was launched and found to be feasible for MCI participants in subsequent studies [43,103]. The second iteration (iPACES v2) of the neuro-exergame was launched in a three-month RCT, where analyses showed that executive function improved in the MCI participants [15]. The present study aims to add evidence to support the use of the iPACES neuro-exergame as an interactive physical and mental exercise intervention to induce cognitive improvements in the MCI population. 

### 1.5. Single-Bout Exercise Intervention and Cognitive Performance

The growing field of various exercise interventions has primarily focused on studying the effects in long-term settings. There is a lack of research surrounding the efficacy of a single session of a physical exercise intervention. Meta-analyses have shown that a single bout of exercise (i.e., less than 60 min) can provide a noticeable improvement in executive function [41,118,119]. Studies have shown that a single bout of exercise can improve cognitive performance in children and younger adults [120,121]. Early studies compared cognitive performance in older adults to that in younger adults and found that acute bouts of aerobic exercise significantly improved executive function and cognitive performance in older adults [118,122]. Many researchers have studied the effects of single-bout exercise on the domains of working memory and inhibitory control [118,123]; additionally, researchers have examined the impact on memory [124,125]. New research looks at how a single-bout exercise session affects different components of executive function, such as cognitive flexibility and task-switching ability [126]. Additionally, a few studies have explored the effects of a single bout of exercise on biomarkers associated with neural growth. An acute bout of aerobic exercise has been shown to increase the serum levels of BDNF, a neurotrophic factor correlated with neural growth [127,128]. Researchers evaluated the effects of a single exercise bout of interactive exergaming in a pilot study of the iPACES v1, and analyses found a significant increase in executive function, represented by increases in performance on neuropsychological tasks, among older adults post-intervention [17]. This study is a partial replication and extension, expanding the investigation to include potential biomarkers linked to cognitive benefits.

### 1.6. Neurobiological Mechanism and Markers of Cognitive Changes

It has been hypothesized that an interactive exergaming intervention can produce greater cognitive benefits due to the synergistic interaction between physical exercise and cognitive stimulation. The research focuses on connecting mental processes to the physiological pathways triggered by exercise, as revealed by measurable changes in particular biomarkers, such as BDNF and IGF-1 [103]. Multiple neurobiological mechanisms are activated by physical exercise, including the regulation of the production and uptake of certain central and peripheral growth factors, which promote synaptic plasticity and improvements in cognition [45]. 

The following is a summary of the current understanding of the role of several of these factors as they relate to cognitive function and dementia. For example, physical exercise has been shown to yield increases in BDNF, which is linked to enhanced plasticity [45,88,129,130,131,132,133,134]. BDNF has been implicated as a neuroprotective growth factor, and it is part of a cascade that promotes neuronal plasticity [45]. However, the use of BDNF as a biomarker attributed to exercise has been quite controversial, with inconsistencies reported in various studies [135]. Insulin-Like Growth Factor (IGF-1) has been discovered to play a variable role in the exercise-induced mechanism that promotes neuroplasticity [45,132,136,137,138]. IGF-1 has been shown to increase with exercise [127,136,139,140,141], and decreased IGF-1 has been associated with decreased cognition [142,143]. BDNF and IGF-1 participate in intertwining pathways that ultimately lead to neurogenesis and improved learning [45]. 

Many studies have studied cortisol and its role in the stress response mechanism that is triggered by exercise. Cortisol, a glucocorticoid hormone involved in the human stress response [144,145], has been highlighted in individuals with cognitive decline as it can cross the blood–brain barrier and affect pathology and cognition [146,147,148]. Studies show that high concentrations of cortisol are linked to changes in cognition in patients with Alzheimer’s and other dementias [144,149,150,151,152,153]. Prior studies have explored the association between cognitive improvements in MCI and changes in cortisol [154]. It was found that cortisol decreased with improvements in executive function in the MCI population [15]. More research is needed to understand the physiological interaction between the stress response and cognitive improvements in MCI. Additionally, salivary alpha-amylase is another biomarker of the human stress response and has been shown to increase in the face of acute stress, making it a viable biomarker to explore in MCI populations [155]. This study contributes to the limited literature on the role of salivary alpha-amylase in the stress response triggered by physical exercise.

### 1.7. A Priori Hypotheses

Executive function will improve after a single-bout intervention of interactive neuro-exergaming [17].Improvements in cognition will be correlated with changes in the salivary biomarkers as follows:
Cortisol will be negatively correlated [15];Alpha-amylase will be positively correlated [155];IGF-1 will be positively correlated [15].MCI participants will experience a greater improvement in cognitive performance than normative older adults [103].

## 2. Materials and Methods

### 2.1. Participants

Participants were enrolled at multiple sites in the Northeastern United States and the study protocol was approved by the IRBs at the relevant institutions. Older adults, aged 50+, were invited to voluntarily participate in a single bout of neuro-exergaming on the tablet-based interactive Physical and Cognitive Exercise System (iPACES v2). Over two years, participants were recruited from an academic neuropsychology lab, independent senior residences, and a physical therapy center. Various recruitment methods were used, including outreach emails by the administrators of programs, the posting of flyers, and demonstrations of the game at senior living communities.

Criteria used to screen participants included physical or mental impairments that would hinder the participant’s ability to complete or understand the cognitive tests, pedal an under-desk elliptical, or play an interactive memory game. Participants were screened using the Impaired Decision-Making Capacity (IDMC) test, a structured interview that ensures that the appropriate consent can be obtained (IDMC; Veteran’s Health Administration Handbook), and they provided informed consent (if applicable; alternatively, they were co-signed by a surrogate or legally appointed representative per the IDMC).

The participants who completed the single bout were 20 females and 13 males (see Table 1).

The mean age of the participants was 72.8 (SD = 10.7), the mean level of education was 16.1 years (SD = 2.4), the mean body mass index was 25.9 (SD = 3.3), the average cognitive status was in the normative range (24.7, MoCA > 23; with 10 meeting the screening criteria for MCI, MoCA ≤ 23) [156], and one individual self-reported a multiethnic background (see Table 1).

Thirty-three participants completed the 20-min single bout of exercise using the under-desk elliptical and the interactive iPACESv2 game (two participants reported parkinsonism; however, their slight tremors did not impede the neuropsychological exercise or evaluation; see Figure 1 and Table 2).

Of the 33 participants, 27 participants were able to provide complete salivary samples, which were used for the neurobiological marker analysis (see Results). Of the 27 participants, 20 participants were included in the pre/post comparative assessments, while seven participants had to be excluded due to incomplete data (either administration error in recording or due to participant factors such as difficulty completing a test). Thus, a total of 20 participants were able to be included in the final analysis, and, of those, five met the screening criteria for MCI.

### 2.2. Procedures

After participants expressed interest in participating in the study, they were screened by a research associate over the telephone or in person using the IDMC structured interview. A time and location were scheduled for the evaluation, in which participants reviewed and signed an informed consent form. Participants completed a demographic form, including age, height, weight, marital status, computer use, and exercise history. During the evaluation, participants completed a neuropsychological battery before and after the exercise intervention, to understand the change in cognitive performance. Saliva samples were collected before and after the exercise intervention using the passive drool technique, to be stored for Enzyme-Linked Immunosorbent Assay (ELISA) analysis to investigate the biomarker correlations relating the changes in cognition to exercise, as described further below. Upon completion of the intervention, participants were asked to complete an “exit” survey questionnaire in which they evaluated the exergaming intervention and experience.

### 2.3. Measures

#### 2.3.1. Neuropsychological Evaluation

The purpose of this intervention was to investigate the correlation between physical exercise and the improvement of executive function. Previous literature has shown that executive function is the major domain that improves due to exercise and exergaming [12,15,17,41,44,59,67,75,93,103,108,109,115]. Executive function refers to higher-order mental processing that is responsible for the coordination and execution of attention and behavior when faced with multiple stimuli that require working memory, inhibition, and cognitive flexibility [157]. These traits are essential for older adults to maintain independence in later life. A battery of neuropsychological assessments was administered to collect data on various aspects of executive function, including the Stroop task, to assess the changes before and after the single-bout intervention.

#### 2.3.2. Cognitive Measures

The Stroop task (40-item format) is an executive function measure that has been used with high validity and reliability in clinical and research studies [158,159,160,161,162]. In Stroop A, participants stated the colors of blocks printed in line on a page; in Stroop B, participants stated the color words printed in black ink; in Stroop C, participants stated the color of the ink and the color words were printed on the page. In Stroop C, some of the color words were congruent (same color ink as the color word) and some were incongruent (different color ink and word), so this required participants to ignore distractor stimuli and allowed for the analysis of effortful inhibition. In the study, both the paper Stroop and an electronic version of the Stroop through an application called Brain Baseline were administered to each patient [15,159].

The Congruent Correct–Incongruent Incorrect metric (CCII; [15,163]) is an executive measure scaling technique that was applied to the electronic Brain Baseline Stroop task. The CCII metric is the percentage of correct congruent responses minus the percentage of incorrect incongruent responses (II; [15]). The objective of the CCII metric is to measure the strength of mental processing by comparing the ability to correctly respond to easier stimuli with incorrect performance when responding to different stimuli. The electronic Brain Baseline Stroop data were used to compute the CCII metric and yielded proportions ranging from −1 to 1.

The Alzheimer’s Disease Assessment Scale (ADAS) Word Recall Task [164] is a verbal memory task used to evaluate short-term memory through immediate recall [165] and long-term memory through delayed recall [166]. In the first three trials, participants verbally read a list of 10 words displayed to them on cards with the following immediate recall. After completing 10 min of another neuropsychological task, participants were asked to recall the list of words again for delayed recall. Different word lists were used for the pre and post assessments. This task was conducted using the Quizlet app on the iPad.

The Montreal Cognitive Assessment (MoCA; [167]) was used to assess the cognitive function of the participants. The test assesses multiple cognitive domains via delayed word recall, a clock drawing task, digit repetition, color trails, and more. This test has been validated and is commonly used as a cognitive screen in clinical research [167]. The scores can be used to categorize participants within various dementia to normative ranges, where scores above 26 are normal, scores below 17 represent an Alzheimer’s-type dementia, and scores between these values represent an MCI condition [167]. However, a recent meta-analysis of 304 studies showed that a cutoff of score of 23 (MCI < 23) had better diagnostic accuracy [156]. An electronic version of the MoCA was used on the iPad.

#### 2.3.3. Biological Measures

Saliva samples were collected from all participants that were willing and able to offer saliva through passive drool before and after the single-bout intervention. The samples were placed on ice at −80 °C and then analyzed in triplicate for concentrations of cortisol (Cohen Biochemistry lab protocol), alpha-amylase (alpha-amylase saliva ELISA, BioVendor, Brno, Czech Republic), and IGF-1 (Human IGF1 SimpleStep ELISA Kit, Abcam, Cambridge, UK). Protein concentrations in the saliva samples were measured using the bicinchoninic (BCA) assay (Pierce™ BCA Protein Assay Kit, Thermo Fisher, Waltham, MA, USA). Measurements of the salivary biomarkers were normalized to the total protein concentrations.

### 2.4. Intervention

The interactive physical and cognitive exercise neuro-exergame was designed by the Healthy Aging and Neuropsychology Lab at Union College to target executive function and was initially programmed for a touch-screen PC tablet (iPACES v1) [17]. Version 2 (iPACES v2) of the game used in this study, called Memory Lane, was created through a collaboration between the lab and a software company, 1st Playable Productions in Troy, NY, to improve the playability and to create an iOS app for the iPad [15]. To complete the neuro-exergaming interactive operation and track the necessary vitals for analysis, multiple Bluetooth-enabled devices were integrated into the software: a heart rate monitor ring on the finger (Hammacher Schlemmer Lifewell); a cadence meter to track the pedaling motion (Polar Global Cadence Sensor Bluetooth Smart), and an under-desk elliptical pedaler (Stamina 55-1610 InMotion E1000 Elliptical Trainer).

The iPACES intervention using the Memory Lane game required participants to use an iPad Air 2, an under-desk elliptical, and a heart rate monitor ring. The iPACES Memory Lane neuro-exergame (v2) was played on the iPad [15]. The heart rate ring was used to monitor the heart rate so that participants could reach their target heart rate during their single bout of exercise [168]. The information taken from the memory game was their total score, minutes pedaled, and average heart rate.

The iPACES Memory Lane game was designed to model a real-life scenario along with the intention of testing and challenging executive function by the simulation of riding a bike along a virtual bike path to complete a list of errands [15]. Participants used the iPad as a steering wheel by tilting the device left and right to “steer” through a virtual bike path. Participants were asked to complete a 20-min exercise session, excluding 1–3 min at the beginning to assimilate to the game. Participants were given instructions for the game, starting with how to follow and steer through a simulation of a virtual bike down a road. When the participants pedaled the under-desk elliptical, the virtual bike moved on the screen at the speed at which the participants were pedaling. The cognitive aspect of the game involved a word list memory task in which participants were instructed to remember words presented to them at the beginning of the path. Within the game, participants navigated through forks in the virtual road, where they had to choose the correct destination based on a list of locations (from the words on the list) presented at the beginning. This task targets the executive function abilities; therefore, the iPACES game is classified as a neuro-exergame [44]. When the participants correctly followed the order of locations, they were instructed to follow the bike path in reverse order. The participants could monitor the total time that they exercised, their total distance traveled, and their current heart rate on the screen. Participants were alerted if they chose an incorrect word. After the intervention, participants completed questionnaires regarding performance during and after the exercise, followed by the second set of neuropsychological assessments.

## 3. Results

### Description of Results

Paired *t*-tests revealed significant changes in salivary alpha-amylase (*p* = 0.01) and executive function, measured by the CCII ratio (*p* = 0.04). A repeated-measures ANOVA was conducted to measure the change in alpha-amylase and executive function, measured by the CCII ratio, in the MCI and normative groups. A significant improvement was found in salivary alpha-amylase and the CCII measure of executive function after the single-bout iPACES intervention [*F*(2,17) = 9.09, *p* = 0.002]. A univariate ANOVA was conducted to measure the change in salivary alpha-amylase and executive function, measured by the CCII ratio, in the MCI and normative groups. A greater increase in salivary alpha-amylase was found in MCI participants compared to normative participants [*F*(1,17) = 13.8, *p* = 0.002] see Figure 2.

In addition, a greater increase in executive function, measured using the CCII ratio, was found in MCI participants compared to normative participants [*F*(1,17) = 4.8, *p* = 0.04], see Figure 3.

Furthermore, a negative correlation was found between the change in salivary alpha-amylase and MoCA, with age as a covariate (*r* = −0.64, *p* = 0.002).

## 4. Discussion

This study is a replication and extension of an earlier feasibility trial that explored whether a single bout of interactive physical and cognitive exercise can yield enhanced cognitive benefits compared to independent physical or mental exercise [17]. The synergistic effects of aerobic exercise and cognitive training have been explored in the older adult population with MCI, and this population has previously demonstrated improvements in short-term and long-term neuropsychological abilities [15,17,44,82,102,103]. The current results were derived from 20 older adults who completed assessments before and after a single bout of pedaling-to-play the iPACES neuro-exergame. The purpose of this study was to explore the cognitive and neurobiological effects of a single session of a combined interactive physical and mental exercise intervention, pedaling an under-desk elliptical that controlled progress in a video game. A battery of neuropsychological tests was used to assess changes in executive function, and salivary samples were used to measure changes in neurobiological markers. The results affirm the a priori hypothesis, revealing a significant change in salivary alpha-amylase and executive function, measured using the CCII ratio, amongst the entire group. These findings are important given the varied literature regarding the cognitive effects of a single bout of exercise and provide further evidence of the possible utility of exergaming for MCI patients. The significant increase in salivary alpha-amylase is an intriguing finding, as it provides further evidence that exercise, herein neuro-exergaming, activates the noradrenergic system, which, to our knowledge, has only been reported in one other previous study [155]. The noradrenergic response has been linked to cognitive and memory improvements and can be the focus of future interventions [155]. However, to our knowledge, this is only the second paper studying alpha-amylase in this way. Neither the Segal et al. (2012) [155] nor this study has determined the physiological mechanisms underlying the cognitive changes. Furthermore, this finding reveals that the MCI population may experience stress during the intervention, and additional research is needed to clarify the role of physiological arousal, stress over the challenge, or other factors in elevating alpha-amylase. Continuing research and development in iPACES is aimed at future versions of neuro-exergaming that will make it an enjoyable experience that allows for the maximum cognitive benefits for both challenged (MCI) and higher functioning, normative participants.

Results revealed a significant increase in executive function, measured using the CCII ratio, and salivary alpha-amylase in both the MCI and normative populations. The increases in both salivary alpha-amylase and executive function were greater in the MCI group compared to the normative group. Additionally, the increase in salivary alpha-amylase was found to have a significant negative correlation with the MoCA scores, as individuals with lower MoCA scores were found to experience the greatest change in salivary alpha-amylase. This suggests that the participants with lower cognitive baselines were more likely to experience increases in cognitive markers, evidence for an increase in cognition. This evidence supports the idea of interventions targeting the MCI/intermediate stage of cognitive decline, because the MCI population can experience improvements in cognitive abilities. This phenomenon is possible because those with MCI may still possess the physiological components to obtain the benefits of the activation of the exercise-induced mechanisms that promote neuroplasticity. Furthermore, future research should include measurements of the neurotrophic and neuroprotective growth factors, such as BDNF and VEGF, that are implicated in the mechanism that promotes neural growth [45], to verify that neurogenesis takes place after a single bout of the iPACES intervention.

An important strength in this study was the use of salivary biomarkers as indicators of cognitive improvement. Different biomarkers have now been implicated in several intertwined neurobiological mechanisms that appear to be activated by physical exercise [45], but further research is needed to determine the role of physical, cognitive, and other factors in the process. The ability to quantitatively measure various salivary biomarkers that suggest cognitive improvements will be beneficial in future exergaming interventions because it is non-invasive and simple, compared to serum-based or cerebrospinal biomarkers. Exergaming interventions have evolved to facilitate greater compliance in the MCI/older adult population, as seen in the switch from the stationary bike “cybercycle” [43,44] to an in-home, portable neuro-exergame [17,103]. A practice such as the spinal tap for cerebrospinal fluid to determine beta-amyloid protein levels may not be practical in the older adult population [169]; therefore, the practice of measuring salivary biomarkers will be important in further exergaming interventions. Classical Alzheimer’s biomarkers such as beta-amyloid have been detected using salivary analysis [170,171,172,173]. There is immense potential for further studies to identify and quantitatively determine certain biomarkers that could allow researchers to construct a framework for the broad neurobiological mechanisms. Furthermore, more studies are exploring biomarkers in serum, cerebrospinal fluid, and imaging in patients with MCI, such as microRNA [174]; thus, it would be beneficial to expand the biomarker analysis in future trials to study the exergaming-induced effects.

Another important strength in this study was the neuro-exergame utilizing the iPACES platform, where an iPad can be paired with an under-desk elliptical. The video game component of the iPACES platform provides adequate motivation and attention for older adults to achieve the recommended levels of exercise necessary to improve cognitive health [15,103]. One purpose of the game is to model a real-life, naturalistic situation of a journey in one’s neighborhood to complete a set of errands and then retrace one’s path to return “home,” and it models the independence that is often eroded by progressive cognitive decline. The concept of this game raises the question of how the cognitive benefits of the video game cognitive training manifest in the participants, and, if it can be generalized to “real life”, whether we can conclude that they can maintain or regain qualities diminished due to underlying Alzheimer’s disease or related dementias. Further research in a larger cohort of participants is needed to understand how a long-term neuro-exergame intervention might affect activities of daily living, as this would shed light on how interactive exergaming targeting the executive function domain can affect one’s overall quality of life and cognitive trajectory. A larger, long-term trial might affirm that interactive neuro-exergaming can produce enhanced cognitive benefits and slow the progression of cognitive decline.

One limitation of the study was the lack of reliability of the technology to measure the heart rate reliably. Data were not obtained for the heart rate measurements because the heart rate device did not function consistently to produce continuous heart rate measurements. Prior research has shown that when participants reach their target heart rate, they can maximize the benefits of physical activity [168]. The fact that the findings were significant- even if not every participant was able to fully sustain their target heart rate or, by inference, achieve the targeted dose of the neuro-exergame- suggests that even a single bout experienced in a naturalistic way, including such variability, may be potent. In this light, the findings may even be more generalizable to a naturalistic environment wherein such variability would be expected. Nonetheless, the more precise measurement and integration of heart rate data would seem likely to boost the significance in future studies. Future trials should include a more reliable system, such as using an axial heart rate system measurement, that can be integrated with the Apple iOS platform to measure adequate vitals throughout the exercise session.

While this study yielded promising findings, some additional limitations might have affected the results. A larger sample size would help to increase the statistical power, identify any outliers and smaller effects, and improve the generalizability of the results when there is a more diverse group that allows for better analysis of the covariates (age, gender, race, education, etc.). Another weakness was that the most common cognitive measure to evaluate the participants for MCI across subsamples was the MoCA, already noted above as a brief screening tool. Additional testing to discern the type and degree of MCI and more specific cognitive challenges could be useful in elucidating more specific benefits or areas for the adaptation of this intervention (e.g., amnestic MCI (aMCI), wherein memory deficits predominate). Future studies should include a breadth of neuropsychological measures to better characterize the patients and a larger sample to allow sufficient additional categorization. Additionally, future studies should pursue a detailed analyses to determine whether there is variability in changes in executive function and salivary markers in response to pedal-n-play exercise as with iPACES. In addition, data regarding the forward/backward learning task in iPACES that targets executive function were not recorded; therefore, future research should quantify these data. This pursuit will provide further insight into the cognitive functioning of the participants, such as how long a word list one can recall in a single-bout session, how many attempts are required for a participant to finish the minimum word list, whether they ever reach maximum word list, and the time taken. Moreover, the participants in this study were primarily European Americans with a generally high level of education and socioeconomic status, so further research must include a more diverse sample, which will help to clarify how iPACES is effective over a broader group of individuals. Furthermore, the limitations of the equipment do not yet allow for the customization of iPACES, and some older adults experienced difficulties with the apparatus and could not fulfill the expected length of the intervention. Future research studies must include various setups with the equipment that allow for adaptability to the participants, which will allow them to perform the intervention in suitable conditions and achieve the full potential of the intervention.

The iPACES exergame has great potential for future use in delaying cognitive decline for MCI patients. While there are pharmacological options to treat cognitive decline, behavioral interventions are an alternative that will revolutionize the treatment of Alzheimer’s disease. Future research should explore the effectiveness of the neuro-exergaming in both short-term and long-term settings and illustrate the neuropsychological and neurobiological outcomes of the intervention. The long-term goal of the iPACES platform is to understand the target amount of neuro-exergaming needed to provide substantial cognitive improvements. The overarching goal of this research should be to uncover and model the exercise-induced neurobiological mechanisms that lead to neural growth, because this would allow clinicians to understand which biological markers to identify when treating the cognitive decline. Future research using the iPACES platform should be undertaken in conjunction with imaging or another form of neurobiological marker measurement, because this would allow for the identification of additional biomarkers indicative of cognitive improvements. The field of non-invasive treatments for the cognitive decline associated with Alzheimer’s disease is growing, and research must continue to clarify the effects of novel behavioral interventions among the MCI population. Ultimately, this may lead to accrued evidence that would encourage clinicians to use behavioral interventions to help ameliorate cognitive impairment.

## 5. Conclusions

This partial replication and extension study revealed that a single bout of a neuro-exergame can induce cognitive improvements and biomarker changes in patients with mild cognitive impairment. Significant increases in biomarkers such as salivary alpha-amylase suggest that this type of exercise can activate neurobiological mechanisms that, in other research, have been linked to neurogenesis, thus providing an indication of a linking mechanism. The improvement in executive function is encouraging and suggests exercise-induced benefits to neurological function. While each of these findings is not entirely new (e.g., the cognitive benefits of exercise are well established, and biomarkers as a possible mechanism are increasingly being revealed [12]), this study provides encouraging new evidence at the intersection of these fields of investigation and takes the research a step further. Older adults, including those with MCI, may obtain a cognitive benefit from an exercise paradigm that is tailored to target a specific cognitive function through an interactive experience, such as executive function, as prospectively designed in iPACES. Furthermore, this may be done more conveniently than in previous protocols, using a simple, portable, and cost-effective exercise intervention that can be implemented in-home for older adults. Interventions such as the iPACES neuro-exergame need additional longer-term investigation to verify that they can be utilized, and further adapted and refined, as a meaningful intervention to curb cognitive decline in patients with prodromal Alzheimer’s disease or related dementias.

## Figures and Tables

**Figure 1 brainsci-13-00844-f001:**
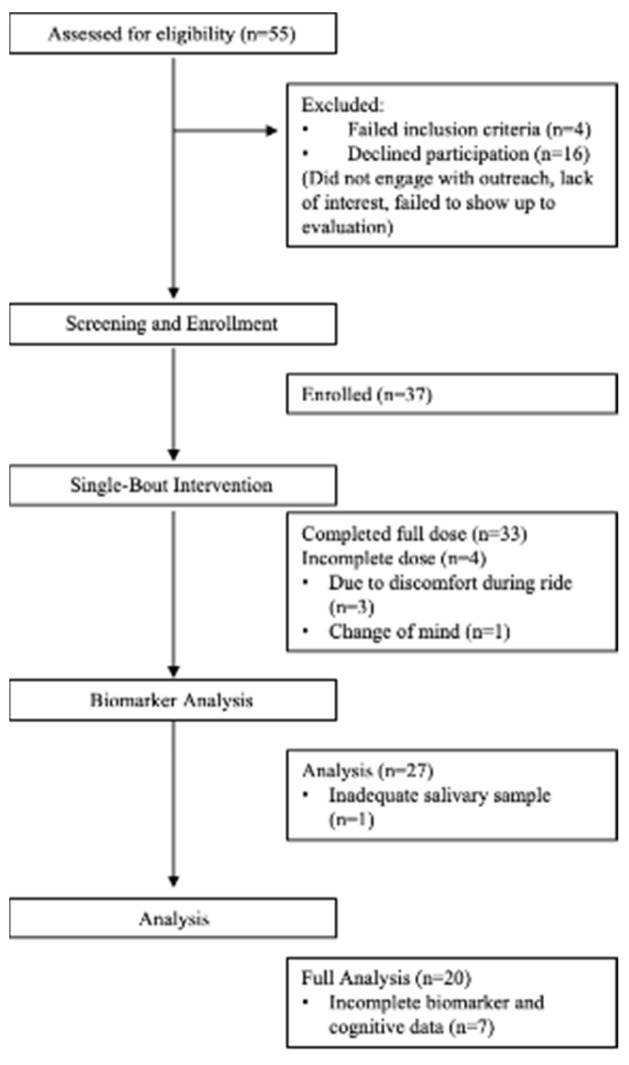
CONSORT flow diagram: enrollment, progress, and analysis throughout the single-bout intervention.

**Figure 2 brainsci-13-00844-f002:**
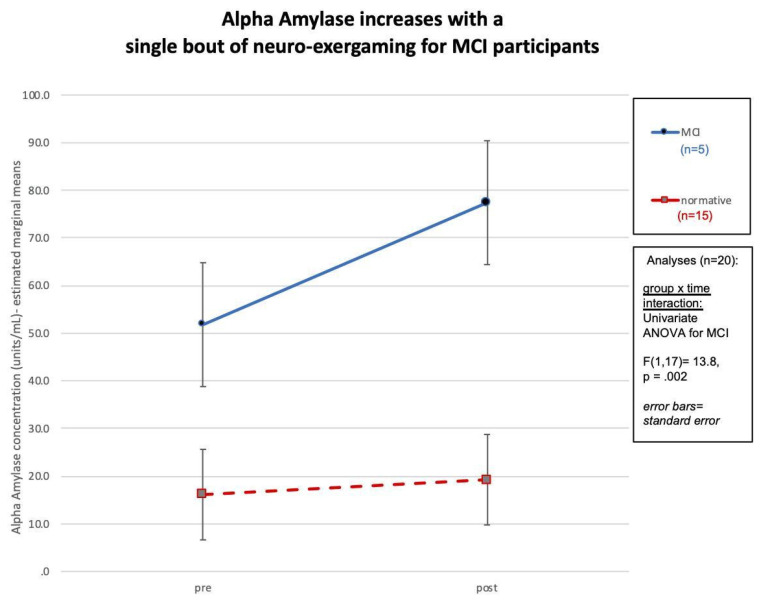
Change in salivary alpha-amylase among the MCI and normative participants, with a significant increase in alpha-amylase among the MCI participants.

**Figure 3 brainsci-13-00844-f003:**
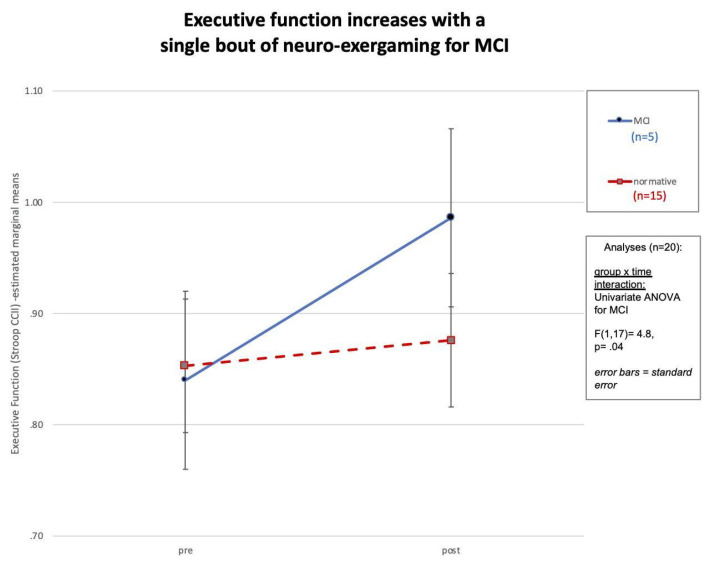
Change in executive function among the MCI and normative participants, with a significant increase in executive function among the MCI participants.

**Table 1 brainsci-13-00844-t001:** Demographics for the iPACES single-bout intervention.

	Completed iPACES Single-Bout		Completed Cognitive & Biomarker Data	
	(*n* = 33)		(*n* = 20)	
Demographics ^a^	X	SD	X	SD
Age	72.8	10.7	70.5	9.0
Education (years) ^b^	16.1	2.4	16.3	2.5
BMI	25.9	3.3	26.0	3.2
MoCA Score	24.7	3.0	25.3	2.6
Gender (% female)	61%		65%	
Race (% White)	98%		98%	
Retiree Status (% retired)	82%		75%	
Marital Status (% married) ^c^	67%		80%	
Self-rated physical activity ^d^	3.0	1.4	3.1	2.5
Past Experience ^e^	0.0		0.0	
Cycling	2.5	0.9	2.5	2.5
Computers	2.8	1.0	3.1	3.1
Videogames	1.1	1.0	1.3	1.3
Motivation	2.5	0.7	2.5	2.5

Abbreviation: iPACES = interactive Physical and Cognitive Exercise System. ^a^ Medical history not formally addressed, one person is each group noted: parksonism, tremor did not affect intervention. ^b^ Many reported 2 and 4 year college degrees; graduate including masters and PhD degrees. ^c^ Does not include widow/widowers. ^d^ Scale 1–5 (Sedentary to aerobic exercise >3 h per week). ^e^ Scale 0–4 (None to lots of experience) (See [17]).

**Table 2 brainsci-13-00844-t002:** Biomarker and cognitive outcomes for the iPACES single-bout intervention.

	Completed iPACES Single-Bout				Completed Cognitive & Biomarker Data			
	(*n* = 33)				(*n* = 20)			
	Pre		Post		Pre		Post	
Measure	X	SD	X	SD	X	SD	X	SD
Executive Function								
Stroop (Brain Baseline)								
Incongruent % correct	0.83	0.25	0.89	0.19	0.91	0.15	0.93	0.13
Congruent % correct	0.92	0.17	0.97	0.07	0.94	0.14	0.98	0.07
CCII	0.75	0.37	0.85	0.26	0.85	0.25	0.90	0.20
Stroop (Paper)								
Stroop A time	24.94	6.96	24.60	8.11	22.65	5.17	22.54	6.36
Stroop B time	19.16	6.73	19.20	5.84	17.62	4.17	18.62	5.34
Stroop C time	57.94	32.75	52.58	22.51	48.75	21.22	46.50	19.29
ADAS-Cog								
Word List (sum trials correct)	18.54	4.37	19.75	3.86	19.89	3.74	21.25	2.99
Word List (delay correct)	6.00	2.85	5.33	3.25	7.25	2.05	6.42	2.75
Biomarkers								
cortisol					1.23	0.81	1.09	1.25
IGF-1					384.82	241.29	441.20	238.31
alpha-amylase					37.72	68.66	33.83	45.67

Abbreviations: iPACES= Interactive Physical and Cognitive Exercise System, ADAS = Alzheimer’s Disease Assessment Scale, CCII = Congruent Correct-Incongruent Incorrect Metric, IGF-1 = Insulin-Like Growth Factor.

## Data Availability

The data presented in this study are available on request from the corresponding author. Results were previously presented at the Annual Meeting of the Eastern Psychological Association in 2020. The following is the citation. “Nath, K., Puleio, A., Michielli, M., Voelm, C., Alberts, K., Wall, K., Duff, A., Hanley, C., Rogers, C., Cohen, B. & Anderson-Hanley, C. (2020). Biomarker and cognitive improvements after a single bout of Interactive Cognitive and Physical Exercise (iPACES v2.75): Results of a multi-site pilot study” [175].

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
