# Peer review of "Brain Health Indicators Following Acute Neuro-Exergaming: Biomarker and Cognition in Mild Cognitive Impairment (MCI) after Pedal-n-Play (iPACES)"

_brainsci, 2023, doi:10.3390/brainsci13060844_

Round 1

Reviewer 1 Report

  • Summary: The authors seek to evaluate a previously created combination exercise and cognitive training intervention for older adults at risk for cognitive decline. The design is creative and potentially very useful in the broader community. This particular paper seeks to evaluate the effects of a single bout of training on 20 participants (5 with MCI).
  • Major Strength: The background and review for this manuscript was extensive and excellent. It was almost as if this paper was seeking to be a review (with 182 references) rather than a manuscript reporting on experimental results. The authors use this review to make a case for a specific hypothesis about noradrenergic function and MCI, and seek to test that with salivary measures.
  • Major Weakness: However the intention of the authors did indeed appear to be to report on experimental results. Those were unfortunately thin, however.
  • Specific concerns:
    • MCI is a phenomena that covers a broad range of function. Yet, there is no description offered to tell the reader about the functioning of these MCI participants. If all that was used to categorize participants was MOCA, there is much more to characterizing these participants to have a sense of how the stage of MCI is related to performance.
    • 13 participants from the full sample were excluded. It appears that only 27 completed full salivary samples, and presumably the remaining 7 did not complete the cognitive battery? The authors stated that all participants could complete the cycling. More description would help us understand why a third of enrolled participants did not complete this single-session study.
    • Given that only 10 of the 33 participants were reported to have scored in the MCI range and only 20 did the full study. The authors did not report how many MCI participants were remaining in the sample in the participants section. From the results, it is clear that (again, not clear why) half of the MCI participants were lost.
    • The results section is really thin. How did participants perform on average on the cognitive aspects of the intevention? The results only show correlations with salivary markers but not the performance itself.
    • No data on HR or cadence was offered, although based on the review, it is reasonable to assume that those achieving higher heart rate levels or cadence may have better effects on the intervention. Was this the case?
    •  
    • Minor concerns: The first statement of the discussion appears to be copied from something like author instructions. "Authors should discuss the results and how they can be interpreted from the per-spective of previous studies and of the working hypotheses." I would recommend deleting.
    • Minor concern: Authors discuss drawbacks about fingertip HR monitoring. Please note that with frequent hand movements as will be required for iPACES, a watch may be no better. Axial HR measurements are the best especially for HRV measures to minimize artifact.    

Author Response

Please see the latest attachment. 

Reviewer 2 Report

The authors present a novel one-shot intervention study. It is interesting to see the efficacy of neuroexergaming over just one intervention session! Suhc studies can pave the way for more studies that examine the effect of such interventions. I encourage the authors to carefully proof read the manuscript again. I found a few lines at the start of discussion (Pg 11)  to be form a template, rather than a summary of study goal/finding.

Reviewer 3 Report

This thesis is well structured and gives a good overview of the subject matter. In terms of content, however, the findings are not new. In summary, the result is that exercise improves motor and cognitive abilities.

Reviewer 4 Report

This topic in interesting, but some points need to be revised:

- Introduction section is too long. Reduce it.

- Change "A Priori Hypotheses" to purpose of the paper. Revise it.

- "Many researchers study the effects of single bout exercise on the domains of working memory and inhibitory control [128,133]." Correlation with  long term memory should be discussed.

- Improve discussion section with some historical part. These papers are very interesting. Include these: --- doi: 10.1016/j.wneu.2015.06.041 -- doi: 10.1016/j.23.03.001 --  doi: 10.3389/fpsyg.2021.686005 -- doi: 10.1159/000494959 -  Have these awarenesses changed?

- "Improvement in a cognitive domain such as executive function found herein, is encouraging and suggest exercise-induced benefits to brain function." improve this part in conclusion section.

- "Furthermore, critique of the literature has found further limitations including reporting, generalizability, and more that weaken the enthusiasm surrounding the findings [68, 74-78]. Given the controversy regarding the cognitive benefits resulting from cognitive training alone" What does this paper can add more? improve and revise

Round 2

Reviewer 4 Report

Good.